# Diagnosis of COVID-19 in Patients with Negative Nasopharyngeal Swabs: Reliability of Radiological and Clinical Diagnosis and Accuracy versus Serology

**DOI:** 10.3390/diagnostics11030386

**Published:** 2021-02-25

**Authors:** Antonio Mirijello, Michele Zarrelli, Giuseppe Miscio, Angela de Matthaeis, Pamela Piscitelli, Cristiano Matteo Carbonelli, Annabella Di Giorgio, Michele Inglese, Gianluca Libero Ciliberti, Carmen Marciano, Cristina Borelli, Doriana Vergara, Giulia Castorani, Grazia Vittoria Orciulo, Lazzaro Di Mauro, Massimo Carella, Annalisa Simeone, Massimiliano Copetti, Maurizio Angelo Leone, Salvatore De Cosmo

**Affiliations:** 1Unit of Internal Medicine, Department of Medical Sciences, IRCCS Casa Sollievo della Sofferenza, 71013 San Giovanni Rotondo, Italy; angeladematthaeis@gmail.com (A.d.M.); pamela.piscitelli@gmail.com (P.P.); c.carbonelli@operapadrepio.it (C.M.C.); nglmhl@yahoo.it (M.I.); gian.ciliberti@gmail.com (G.L.C.); s.decosmo@operapadrepio.it (S.D.C.); 2Unit of Neurology, Department of Medical Sciences, IRCCS Casa Sollievo della Sofferenza, 71013 San Giovanni Rotondo, Italy; m.zarrelli@operapadrepio.it (M.Z.); ab.digiorgio@operapadrepio.it (A.D.G.); m.leone@operapadrepio.it (M.A.L.); 3Unit of Laboratory Medicine, IRCCS Casa Sollievo della Sofferenza, 71013 San Giovanni Rotondo, Italy; g.miscio@operapadrepio.it (G.M.); l.dimauro@operapadrepio.it (L.D.M.); 4Scientific Research Department, IRCCS Casa Sollievo della Sofferenza, 71013 San Giovanni Rotondo, Italy; c.marciano@operapadrepio.it (C.M.); m.carella@operapadrepio.it (M.C.); 5Unit of Radiology, IRCCS Casa Sollievo della Sofferenza, 71013 San Giovanni Rotondo, Italy; cristinaborelli@hotmail.it (C.B.); dorianavergara@hotmail.it (D.V.); giulia.castorani90@gmail.com (G.C.); graorc@gmail.com (G.V.O.); a.simeone@operapadrepio.it (A.S.); 6Unit of Biostatistics, IRCCS Casa Sollievo della Sofferenza, 71013 San Giovanni Rotondo, Italy; m.copetti@operapadrepio.it

**Keywords:** interstitial pneumonia, new coronavirus Sars-Cov-2, serologic test, sensitivity, specificity, chest CT-scan

## Abstract

Background: The diagnosis of Coronavirus disease 2019 (COVID-19) relies on the positivity of nasopharyngeal swab. However, a significant percentage of symptomatic patients may test negative. We evaluated the reliability of COVID-19 diagnosis made by radiologists and clinicians and its accuracy versus serology in a sample of patients hospitalized for suspected COVID-19 with multiple negative swabs. Methods: Admission chest CT-scans and clinical records of swab-negative patients, treated according to the COVID-19 protocol or deceased during hospitalization, were retrospectively evaluated by two radiologists and two clinicians, respectively. Results: Of 254 patients, 169 swab-confirmed cases and one patient without chest CT-scan were excluded. A total of 84 patients were eligible for the reliability study. Of these, 21 patients died during hospitalization; the remaining 63 underwent serological testing and were eligible for the accuracy evaluation. Of the 63, 26 patients showed anti-Sars-Cov-2 antibodies, while 37 did not. The inter-rater agreement was “substantial” (kappa 0.683) between radiologists, “moderate” (kappa 0.454) between clinicians, and only “fair” (kappa 0.341) between radiologists and clinicians. Both radiologic and clinical evaluations showed good accuracy compared to serology. Conclusions: The radiologic and clinical diagnosis of COVID-19 for swab-negative patients proved to be sufficiently reliable and accurate to allow a diagnosis of COVID-19, which needs to be confirmed by serology and follow-up.

## 1. Introduction

The diagnosis of Coronavirus disease 2019 (COVID-19) stands on the positivity of real-time reverse-transcriptase-polymerase-chain-reaction (RT-PCR) from nasopharyngeal swab [1,2], in addition to laboratory, clinical-epidemiological, and radiologic findings [1]. However, although specificity of RT-PCR is close to 100% [2], sensitivity varies from 33 to 80% [1] depending on several factors, such as time from exposure, accuracy, and adequacy of sample collection. Consequently, 20–67% of nasal swabs may be false negatives [1,3]. 

Radiological aspects of COVID-19 are variable and rely on the sensitivity of the diagnostic method [1]. Generally, initial CT findings of COVID-19 pneumonia are represented by bilateral ground glass opacities (GGO), with a predominantly subpleural location. In a more severe stage of disease, intralobular reticulations superimposed to GGO (“crazy paving” pattern) and consolidations can be present. Several days after disease onset, CT scan can show linear opacities and perilobular consolidations, suggesting an organizing pneumonia, which is a common reaction to lung injury [4,5,6].

Chest CT-scans significantly contribute to the early identification of patients with lung involvement, increasing the probability of a correct diagnosis in patients with an initially negative swab by about 15% [1,7]. Antibodies against SARS-Cov-2 become detectable approximately two to three weeks after infection [1,2]. Although useful for epidemiologic purposes or to confirm previous infection particularly in patients with negative swabs [8], their role in the management of acute patients is unclear. 

At present, the need for urgent treatment and respiratory isolation for a high number of patients affected by respiratory failure due to COVID-19 requires a modulation of hospital activities in order to provide an adequate treatment and, at the same time, to limit the risk of viral transmission to other patients and hospital staff [9]. In this scenario, many efforts are aimed to the development of simple, cheap, and widely available diagnostic tools for the early identification of COVID-19 patients [10], and for differentiating COVID-19 from other acute diseases, particularly when swab results are not available (i.e., shortage of reagents, delays in analyses) or inconclusive. With this regard, the evaluation of markers such as the Neutrophil-to-Lymphocyte Ratio (NLR) [11] and the Model for possible Early COVID-19 Recognition Score (MECOR score) could help in differentiating COVID-19 from community acquired pneumonia [12].

However, data concerning the diagnosis of COVID-19 in patients with negative RT-PCR are lacking. In a sample of hospitalized patients with suspected COVID-19 and multiple negative molecular nasopharyngeal swabs, we aimed to ascertain the reliability of the diagnosis established by radiologists and clinicians and the accuracy of their diagnoses versus serology.

## 2. Materials and Methods

### 2.1. Patients

From 3 March to 31 May 2020, a total of 254 patients were admitted to the COVID-19 Unit of our 900-bed research hospital for suspected Sars-Cov-2 infection. Inclusion criteria were: age ≥ 18-year-old and admission to the COVID Unit. The only exclusion criterion was consent denial. Those with at least one positive RT-PCR were considered swab-confirmed cases. Patients with multiple negative RT-PCR and treated according to the COVID-19 protocol or deceased during hospitalization represent the study population. A serum sample was collected after at least 15 days from admission, or at follow-up. Patients’ blood samples were subjected to Sars-Cov-2 serology with Immunochromatographic assay (Technogenetics—KHB Group), starting from April 2020 and following manufacturer’s protocol. Briefly, 10 μL of serum were added onto the sample loading area followed by 3 drops of sample dilution solution. After no longer than 15 min of incubation, viral IgM- or IgG-containing positive samples could show up both the T line (test) and C line (control); the samples with only C line were scored as negative. In order to validate the method and avoid false results, patients’ sera were frozen (−80 °C) and stored until a chemiluminescent microparticle immunoassay (CMIA) was available (late May 2020), and it was used for the qualitative detection of IgM/IgG antibodies on Abbott’s Architect platform.

The study was approved by the Ethics Committee (CSS-COVID-19 Group, num. 46/2020, 8 April 2020). Informed consent was obtained from all patients. For deceased patients and for those unable to give their consent, the Ethics Committee only allowed for collecting data from the clinical records. 

### 2.2. Reliability

Two radiologists, blinded to the clinical records, independently revised chest CT-scans and classified CT findings into three diagnostic categories (0 = negative; 1 = typical; 2 = indeterminate) [13]. The first category—“negative for COVID-19”—included both normal CT and CT features ascribable to alternative diagnoses. The second category—“typical for COVID-19”—included typical CT patterns (e.g., classical bilateral and subpleural ground glass opacities, “crazy paving” pattern, multifocal ground glass opacities, consolidations with association of septal thickening) [4,14]. The third category—“indeterminate for COVID-19”—included possible overlaps with other entities, or atypical features that may also suggest alternative diagnoses [13]. Similarly, two clinicians revised clinical records and decided on the diagnosis of COVID-19 (0 = likely; 1 = unlikely). 

### 2.3. Accuracy

Both radiologists and clinicians were blinded to serologic results, which were our gold standard. We considered three categories of diagnostic certainty for radiologists: high level of certainty: both radiologists made a diagnosis of COVID-19 (1-1); low level of certainty: diagnosis was made by just one of them (0-1, 1-0, 1-2, 2-1); non-COVID-19: both scored negative (0-0) or indeterminate (2-2). Similarly, we considered three categories for clinicians’ evaluation: high level of certainty (1-1); low level of certainty (1-0, 0-1); non-COVID-19 (0-0). 

### 2.4. Statistical Analysis

Inter-rater agreement was evaluated as both unweighted kappa statistic (κ) and simple percent agreement, according the methodology described by Fleiss [15]. Kappa values of <0.2 were considered poor, 0.2–0.4 fair, 0.41–0.6 moderate, 0.61–0.8 substantial, and >0.8 almost perfect, according to Landis & Koch [16]. We also calculated the agreement between radiologists and clinicians, based on their evaluation stratified by three categories of diagnostic certainty. 

All estimates are reported along with their 95% confidence intervals (95% CI).

Sensitivity, specificity, positive predictive value (PPV), and negative predictive value (NPV) were calculated, along with their 95% CI, considering serology as the gold standard and the radiologists’ and the clinicians’ evaluations—taken separately—as diagnostic tests. 

All analyses were performed using R version 4.0.2 (package epiR) [17].

## 3. Results

Of 254 patients with suspected infection, 169 swab-confirmed cases were excluded. The remaining 85 patients showed multiple negative RT-PCRs. One CT-scan was unavailable. Consequently, 84 patients were eligible for the reliability study. A total of 21 patients died during hospitalization; the remaining 63 underwent serology and were eligible for the accuracy evaluation. Of these, 26 (41.3%) showed the presence of antibodies (antibody-confirmed cases), while 37 (58.7%) did not (unconfirmed cases).

### 3.1. Reliability

The inter-rater agreement was “substantial” between radiologists (Table 1a), “moderate” between clinicians (Table 1b), and “fair” between radiologists and clinicians (Table 1c). 

The overall radiologists-clinicians agreement was “fair” both for antibody-confirmed cases (percent agreement 69.2%, 95% CI: 48.1–84.9; kappa 0.246, 95% CI: −0.107–0.600) and unconfirmed cases (percent agreement 56.8%, 95% CI: 39.6–72.5; kappa 0.245, 95% CI: −0.009–0.600). There was no radiologists-clinicians agreement for deceased patients (percent agreement 42.9%, 95% CI: 22.6–65.6; kappa 0.000, 95% CI: −0.304–0.304).

### 3.2. Accuracy

Table 2 shows the accuracy of the radiologists’ (A) and the clinicians’ (B) judgments versus serology (gold standard). The sensitivity of the radiologists’ evaluation only increased by 2% when diagnoses having a low level of certainty were also considered, at the cost of a considerable decrease in specificity (18%) and PPV (16%). The sensitivity of the clinicians’ evaluation did not change, whereas specificity decreased by 28%, and PPV by 23%.

## 4. Discussion

Patients with repeated negative swabs represent a considerable part of those hospitalized for suspected COVID-19 [1]. In these patients, the diagnosis is challenging and it mostly relies on radiologic and clinical judgment. Although deeper biological samples (e.g., sputum, tracheal-, or brocho-alveolar fluid) can be used to increase the sensitivity of naso-pharyngeal swabs [18], these techniques are not routinely performed. Moreover, confirming as well as ruling-out a diagnosis of COVID-19 is of paramount importance for a correct allocation of patients [9], in order to prevent intra-hospital viral shedding [12]. 

This is the first study to report on the reliability and accuracy of radiologic and clinical evaluations in this particular subset of patients from a real-world setting. We found a higher agreement between radiologists than between clinicians, being substantial and moderate, respectively, according to Landis and Koch classification [10]. 

The substantial agreement among radiologists for chest CT categorization is in accordance with a recent study by Debray and coworkers [19]. In this report, authors observed a good performance of radiologists for predicting COVID-19 pneumonia using a similar categorical assessment scheme. However, it should be emphasized that CT evaluation of patients hospitalized for suspected Sars-Cov-2 infection with multiple negative RT-PCRs may be influenced by the high prevalence of COVID-19 disease during the pandemic, with a possible increase of false positive cases.

The moderate agreement observed among clinicians should be attributed to the typology of the reviewed data (clinical charts) and to the fact that clinicians deal with a larger and discrepant amount of information than radiologists [20]. With this regard, patients carrying a complex medical history with potentially overlapping chronic diseases (i.e., heart failure, hypertensive cardiomyopathy, pulmonary involvement due to rheumatologic disorders, chronic obstructive pulmonary disease, etc.) could show a “blurred image” of their acute state (COVID-19).

We found an even lower agreement between radiologists and clinicians, both in antibody-confirmed and unconfirmed cases, with no agreement for deceased patients. That was expected, at least in part, due to the greater amount of information that the clinicians had to deal with [20].

Both radiologic and clinical evaluations were accurate when compared to serology, showing high sensitivity and specificity, good NPV, and poor PPV (Table 2a,b). The addition of diagnoses with a low level of certainty reduced specificity and PPV, due to an increase of false positives [19]. 

Current management of patients with suspected COVID-19 and negative swabs is uncertain, due to the lack of guidelines [21]. A recent Italian retrospective analysis comparing confirmed versus unconfirmed COVID-19 cases did not show substantial differences in clinical (e.g., symptoms, incubation periods) and radiological findings or mortality between the two groups. However, confirmed cases developed acute respiratory distress syndrome (ARDS) more frequently [22].

Our data confirm that chest CT scans can help early identification of most patients with lung involvement, even in the absence of swab results [7,23], although they are not sufficient to conclusively rule-in or rule-out the presence of COVID-19 [24,25]. Likewise, clinical evaluation can also identify most COVID-19 patients on the basis of suggestive, even if less specific, features typical of the coronavirus infection. In other words, chest CT scans and clinical evaluation are able to correctly identify most of the COVID-19 patients and to rule-out most unaffected patients. 

Limitations of our study include its small sample size, the possible non-generalizability of our results to other settings, and the choice of serology as the gold standard to confirm a previous infection. 

## 5. Conclusions

In conclusion, the radiologic and clinical evaluations are sufficiently accurate and reliable to allow a diagnosis of COVID-19 in patients with repeated negative swabs. On this basis, we suggest also reporting negative-swab patients in clinical series of COVID-19 patients, provided a clinical-radiologic diagnosis is done.

## Figures and Tables

**Table 1 diagnostics-11-00386-t001:** Agreement for the diagnosis of COVID-19 between the two radiologists (**a**), the two clinicians (**b**), and radiologists vs. clinicians (**c**).

(**a**)							
			**Second Radiologist**				
**First Radiologist**			**0**	**1**	**2**	**Total**	
0 (negative for COVID-19)			23	0	0	23	
1 (typical for COVID-19)			0	27	0	27	
2 (indeterminate for COVID-19)			1	17	16	34	
Total			24	44	16	84	
(**b**)							
			**Second Clinician**				
**First Clinician**			**0**		**1**		**Total**
0 (likely COVID-19)			33		8		41
1 (unlikely COVID-19)			15		28		43
Total			48		36		84
(**c**)							
		**Radiologists**					
		**1/1**	**1/0, 0/1, 1/2, 2/1**		**0/0, 2/0, 0/2, 2/2**		**Total**
**Clinicians**	1/1	18	4		6		28
	1/0, 0/1	7	6		10		23
	0/0	2	7		24		33
	Total	27	17		40		84

(**a**) Substantial agreement: percent agreement 78.5% (95% CI: 68.0–86.4), Kappa 0.683 (95% CI: 0.560–0.806); (**b**) Moderate agreement: percent agreement 72.6% (95% CI: 61.6–81.5), Kappa 0.454 (95% CI: 0.267–0.642); (**c**) Fair agreement: percent agreement 57.1% (95% CI: 45.9–67.7), Kappa 0.341 (95% CI: 0.186–0.496).

**Table 2 diagnostics-11-00386-t002:** Accuracy of radiological (**a**) and clinical diagnosis (**b**) vs. serology

(**a**)							
**COVID-19 Higher Certainty**	**Serology +**		**Serology −**			**Total**	
Radiological 1/1	19		5			24	PPV 0.79 (0.58–0.93)
0/0, 2/2	4		24			28	NPV 0.86 (0.67–0.96)
Total	23		29			52	
	Sens 0.83 (0.61–0.95)		Spec 0.83 (0.64–0.94)				
**COVID-19 lower certainty**							
Radiological 1/1, 1/0, 0/1, 1/2, 2/1	22		13			35	PPV 0.63 (0.45–0.79)
0/0, 2/2	4		24			28	NPV 0.86 (0.67–0.96)
Total	26		37			63	
	Sens 0.85 (0.65–0.96)		Spec 0.65 (0.47–0.80)				
(**b**)							
**COVID-19 Higher Certainty**		**Serology +**		**Serology −**	**Total**		
Clinical 1/1		20		5	25		PPV 0.80 (0.59–0.93)
0/0		2		19	21		NPV 0.90 (0.70–0.99)
Total		22		24	46		
		Sens 0.91 (0.71–0.99)		Spec 0.79 (0.58–0.93)			
**COVID-19 lower certainty**							
Clinical 1/1, 1/0, 0/1		24		18	42		PPV 0.57 (0.41–0.72)
0/0		2		19	21		NPV 0.90 (0.70–0.99)
Total		26		37	63		
		Sens 0.92 (0.75–0.99)		Spec 0.51 (0.34–0.68)			

Sens = sensitivity; Spec = Specificity; PPV = positive predictive value, NPV = negative predictive value.

## Data Availability

Data supporting reported results may be provided on reasonable request.

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
