# Peer review of "Diagnosis of COVID-19 in Patients with Negative Nasopharyngeal Swabs: Reliability of Radiological and Clinical Diagnosis and Accuracy Versus Serology"

_diagnostics, 2021, doi:10.3390/diagnostics11030386_

Round 1

Reviewer 1 Report

The authors performed a study in order to use alternative diagnostic techniques for patients with negative swabs for SARS-CoV-2. The paper is interesting, but some issues should be addressed before considering it for publication:

  1. Introduction: A recent paper on diagnostics that I suggest to cite, developed a mathematical score based on WBC to diagnose covid in asymptomatics patients: DOI: 10.3390/diagnostics10090619 - The findings of this paper should be reported in the introduction (and for a future paper, given the high affinity of criteria the authors can try to externally validate the score). Also, I suggest to report some of the typical radiological findings of COVID-19 pneumonia.
  2. In the methods section the authors should specify the software used for statistical analyses and the p-value for statistical significance.
  3. Discussion: the authors should provide more literature to highlight the significance of their findings.

Author Response

We thank the Reviewer for the time spent in the revision of our article and for the suggestions that improved the quality readability of the paper.

  1. Introduction: As suggested by the Reviewer, we have cited the paper by Sambataro and Colleagues in the Introduction section. Again, we thank the reviewer and we will try to externally validate the score.
  2. Introduction section has now been expanded including CT findings of COVID/19 pneumonia.
  3. Methods section: a specific paragraph for statistical analysis was now written, merging statistical information previously written in the “reliability” and “accuracy” paragraphs. The software used (R software) has now been specified. According to the request to report p-values, we preferred to report 95% confidence intervals because much more informative the p-values, especially in diagnostic studies. 95%CI is a direct and intuitive measure of precision as well as inform also about statistical significance: if the 95%CI contains the null value, the estimate is not statistically significant, otherwise if the 95%CI does not contain the null value, the estimate is statistically significant.
  4. Discussion: the section has now been expanded.

Reviewer 2 Report

I think that the lack of this manuscript is the absence of statistical analysis section.

Please add it and modified results section on the basis of statistic.

Author Response

We thank the Reviewer for the time spent in the revision of our article and for the suggestions that improved the quality readability of the paper.

  1. Methods section: a specific paragraph for statistical analysis was now written, merging statistical information previously written in the “reliability” and “accuracy” paragraphs. See also reply to reviewer 1.